# NOD-like Receptors—Emerging Links to Obesity and Associated Morbidities

**DOI:** 10.3390/ijms24108595

**Published:** 2023-05-11

**Authors:** Sarah Bauer, Lucy Hezinger, Fjolla Rexhepi, Sheela Ramanathan, Thomas A. Kufer

**Affiliations:** 1Institute of Nutritional Medicine, Department of Immunology, University of Hohenheim, 70593 Stuttgart, Germany; 2Department of Immunology and Cell Biology, Faculty of Medicine and Health Sciences, Université de Sherbrooke, Sherbrooke, QC J1K 2R1, Canada

**Keywords:** NLRP3, IL-1β, NOD1, NOD2, NLRP12, NLRC5, HFD, inflammasome, microbiota, insulin resistance

## Abstract

Obesity and its associated metabolic morbidities have been and still are on the rise, posing a major challenge to health care systems worldwide. It has become evident over the last decades that a low-grade inflammatory response, primarily proceeding from the adipose tissue (AT), essentially contributes to adiposity-associated comorbidities, most prominently insulin resistance (IR), atherosclerosis and liver diseases. In mouse models, the release of pro-inflammatory cytokines such as TNF-alpha (TNF-α) and interleukin (IL)-1β and the imprinting of immune cells to a pro-inflammatory phenotype in AT play an important role. However, the underlying genetic and molecular determinants are not yet understood in detail. Recent evidence demonstrates that nucleotide-binding and oligomerization domain (NOD)-like receptor (NLR) family proteins, a group of cytosolic pattern recognition receptors (PRR), contribute to the development and control of obesity and obesity-associated inflammatory responses. In this article, we review the current state of research on the role of NLR proteins in obesity and discuss the possible mechanisms leading to and the outcomes of NLR activation in the obesity-associated morbidities IR, type 2 diabetes mellitus (T2DM), atherosclerosis and non-alcoholic fatty liver disease (NAFLD) and discuss emerging ideas about possibilities for NLR-based therapeutic interventions of metabolic diseases.

## 1. Introduction

In the last decades, a strong increase in overweight has been observed in countries with an industrialized Western lifestyle [1]. Overweight describes an excess of body fat and in adults is most commonly defined by a body mass index (BMI) over 25 kg/m^2^ and a BMI of over 30 kg/m^2^ is referred to as obesity. The presence of excessive visceral adipose tissue and the associated increased waist circumference in overweight and obese individuals are linked to various health problems and considered particularly unfavorable. Together with hypertension, hypertriglyceridemia, low high-density lipoprotein (HDL) cholesterol levels and impaired blood glucose control manifesting as hyperglycemia or insulin resistance (IR), the disease pattern is called the metabolic syndrome (MetS), and is a risk factor for the development of type 2 diabetes mellitus (T2DM), atherosclerosis and non-alcoholic fatty liver disease (NAFLD) [2]. The increasing prevalence of obesity and especially the associated morbidities pose a major challenge to health care systems worldwide.

Obesity-associated IR, T2DM, atherosclerosis and NAFLD have been shown to be linked to inflammatory processes. Although clear evidence confirming inflammatory responses as initial triggers of obesity-associated diseases is still lacking, a large body of evidence supports the contribution of inflammatory signaling to the deterioration of obesity-associated morbidities [3]. Increased pro-inflammatory signaling and cytokine release in the adipose tissue (AT) of obese humans and mice fed a high-fat diet (HFD) suggests a key role of adipocytes and AT-infiltrating and -resident immune cells, especially macrophages, as drivers of these inflammatory conditions [4,5,6]. Initiation of the pro-inflammatory signals has been proposed to be mediated by a state termed metabolic endotoxemia, referring to the translocation of microbial and nutritional compounds from the gut into the circulation. This translocation has been shown to be caused by an increase in gastrointestinal permeability upon HFD feeding (leaky gut syndrome) [7,8,9,10,11]. Further, adipocytes in the AT of obese individuals contribute to AT inflammation by dying from hypoxia, as oxygen supply, similar as in tumor tissue, does not accommodate to their extensive increase in size [12]. These dying adipocytes are rapidly surrounded and taken up by adipose tissue macrophages (ATMs), resulting in crown-like structures (CLS) [13,14] and a pro-inflammatory response [15]. Additionally, it has been shown that enlarged adipocytes stimulate collagen synthesis, thus leading to AT fibrosis, which in turn is limiting the expansion and thus lipid storage capacity of adipocytes. This leads to a ”lipid-spillover”, in turn leading to the production of lipotoxic, highly immunogenic molecules, for example, ceramides. By that, AT fibrosis is contributing to AT inflammation, activation of stress pathways and additionally deposition of lipids outside the AT as ectopic fat, for example, in the liver [12,16,17]. All together, these AT-derived pro-inflammatory signals act in a paracrine and autocrine manner, but are also distributed systemically to interfere with the insulin signaling, leading to IR, hyperglycemia and in the end to the development of T2DM [18]. In atherosclerosis, inflammatory responses are primarily initiated by endogenous danger signals present in atherosclerotic lesions and released from ruptured atherosclerotic plaques [19]. Furthermore, in NAFLD, liver inflammation and fibrosis are caused by hepatocellular stress and hepatocyte death due to overnutrition, leading to the production of toxic lipid intermediates, as the liver’s capacity to handle metabolic substrates, primarily carbohydrates and fatty acids, is exceeded. These toxic lipid intermediates induce hepatocellular stress and hepatocyte death, resulting in liver inflammation and fibrosis [20]. Together, these pro-inflammatory signals derived from the respective organs give rise to a state of chronic, low-threshold inflammation in obese individuals, also referred to as sterile or low-grade inflammation.

It has been known for some time that food intake generally leads to alterations in the immune response, manifesting in a mild inflammatory phenotype [21,22,23]. Especially fatty acids have been subject of intense research on and identified as inducers of this postprandial inflammatory status [11,22,24]. Meals high in carbohydrates or a combination of both macronutrients, however, have also been shown to induce a robust postprandial inflammation in healthy subjects, with increased levels of the pro-inflammatory cytokines interleukin (IL) -6 and tumor necrosis factor (TNF) -α, elevated leukocyte counts, increased generation of reactive oxygen species (ROS) and elevated plasma lipopolysaccharide (LPS) concentration [8,9,25,26,27,28,29]. As it has been shown that this postprandial inflammatory reaction is increased in individuals with obesity [27,30,31] and T2DM [32,33] compared to healthy subjects, it was proposed that in a scenario of constant overnutrition or already existing metabolic disturbances, the postprandial inflammation contributes to the development and/or deterioration of metabolic diseases. However, recent evidence suggests a physiological role of this postprandial inflammatory state in healthy individuals. Glucose-driven postprandial increase in macrophage-derived IL-1β, for example, has been shown to be critical for the maintenance of adequate postprandial insulin secretion [34] and ROS, in low to moderate concentrations, have been shown to be beneficial, conferring functions as signaling molecules [35]. Thus, the nature and extend of the postprandial inflammatory state seems to be dependent on the individual metabolic situation, contributing to the maintenance of important physiological processes such as insulin secretion in metabolically healthy individuals, while potentially deteriorating metabolic diseases such as T2DM or obesity.

In addition to impacting metabolic processes, HFD intake also changes the gut microbiome, with decreased abundance of the *Bacteroidetes* phylum [36,37] and increased proportions of the phyla *Firmicutes* [36,37] and *Proteobacteria* [37] being reported in mice. HFD can also increase the species abundance (alpha diversity) in murine fecal samples [38]. Notably, microbial alterations are mainly driven by the HFD feeding and not by the state of obesity itself [37,38]. Interestingly, germ-free (GF) mice are protected from HFD-induced inflammation and IR [39,40] and microbiota transplantation from diet-induced obesity (DIO) mice to GF lean mice led to more fat accumulation compared to transplanting microbiota from lean donors [41]. These data argue for a role of the microbiome in weight gain in obesity. Mechanistically, it was proposed that HFD or Western-style diet (WD), which is rich in simple carbohydrates and saturated fatty acids and low in fiber, increases the abundance of bacterial species able to import and process, and thus extract energy from, simple sugars [41]. In addition to its influence on the gut microbiome composition, HFD has also been shown to disrupt intestinal integrity, leading to increased serum levels of commensal LPS and peptidoglycan (PGN) fragments, thus contributing to endotoxemia and pro-inflammatory responses in peripheral tissues [7,8,9,10,11]. These data highlight that HFD/WD and the gut microbiome mutually impact each other and that these interactions have serious consequences for body weight homeostasis and metabolic inflammation and diseases.

As both, low-grade inflammation in obese individuals and postprandial inflammation present with an increase in circulating immunogenic mediators (LPS, cytokines or ROS), it is not surprising that recent advances show key sensor molecules of the innate immune system, so-called pattern recognition receptors (PRR), to play a central role in the release of pro-inflammatory cytokines and initiation of pro-inflammatory processes in obesity [42]. Deficiency in the main LPS sensor, Toll-like receptor 4 (TLR4), for example, was shown to partly protect female mice from HFD-induced IR and inflammatory gene expression in AT and liver [43]. More recent results show that also proteins of the nucleotide-binding and oligomerization domain (NOD)-like receptor (NLR) family are involved in metabolic inflammatory reactions [44,45]. 

NLRs are a group of cytosolic PRRs that share a common tripartite domain organization consisting of a central ATPase and oligomerization domain (NACHT), mediating the oligomerization and thus activation of NLRs, a variable number of C-terminal leucine-rich repeats (LRRs), responsible for recognizing ligands, and a variable N-terminal effector domain, based on which NLRs can be classified into four sub-families, NLRA, NLRB, NLRC and NLRP (Figure 1). NLRAs possess an N-terminal caspase-activation and recruitment domain (CARD) associated with an acidic transactivation domain (AD); NLRBs contain a baculovirus inhibitor of apoptosis protein repeat (BIR) domain [46,47,48]. No associations with obesity have yet been described for representatives of these two NLR sub-classes, which is why they are not discussed further in this review article. NLRCs carry one or two CARD or CARD-like domains at the C-terminus, associated proteins are NOD1 (NLRC1), NOD2 (NLRC2), NLRC3, NLRC4 and NLRC5. The effector domain of the NLRPs consists of a pyrin domain (PYD), and the corresponding NLRs are designated NLRP1-14 [46,47,48].

NLR proteins confer multiple cellular functions. Some NLRs function as PRRs and recognize components of pathogenic microorganisms (pathogen-associated molecular patterns, PAMPs, also known as microbe-associated molecular patterns, MAMPs), such as PGN, a component of the bacterial cell wall, flagellin or viral RNA [49]. However, NLRs can also react to incorrectly localized or pathologically altered endogenous molecules, so-called danger-associated molecular patterns (DAMPs). Examples include extracellular ATP, crystallized cholesterol or uric acid crystals [19,46,47]. NLRs functioning as bona fide PRRs and inducers of pro-inflammatory responses, for example, are NOD1, NOD2 and NLRP3. NOD1 and NOD2 induce pro-inflammatory responses via the adaptor molecule receptor-interacting serine/threonine-protein kinase 2 (RIPK2) through activation of the nuclear factor ”kappa-light-chain-enhancer” of activated B-cells (NF-κB) upon recognizing their specific ligands [46]. NLRP3, and several other NLRP proteins, form so called ”inflammasomes”, multiprotein complexes consisting of the corresponding NLR, apoptosis-associated speck-like protein containing a CARD (ASC) and pro-caspase-1. Activation of the inflammasome by a two-step process induces cleavage-mediated activation of caspase-1, ultimately resulting in the processing and thus activation of IL-1β and IL-18 [50,51,52,53]. Today, we know that NLRs also confer important functions beyond the initiation of inflammatory processes. NLRP11 and NLRP12, for example, have been shown to regulate rather than initiate immune signaling [54,55], whereas NLRC5 and Class II Major Histocompatibility Complex Transactivator (CIITA) function as transcriptional activators and are the master regulators for the transcription of the major histocompatibility (MHC) class I and class II genes, respectively, thus linking innate and adaptive immune responses [46,56,57]. 

In the context of obesity, low-grade and postprandial inflammation, the physiological relevance of these proteins, their triggers and molecular mechanisms still largely remain elusive. We here summarize the current state of research on NLRs in obesity, low-grade and postprandial inflammation and discuss which signaling events may play a role in the pathogenesis of obesity and its associated morbidities (Table 1). Of the 22 human NLRs known to date [48], six proteins (NLRP3, NLRP6, NOD1, NOD2, NLRC5 and NLRP12) (Figure 1) have been described to be associated with obesity and low-grade inflammation and one (NLRP3) with postprandial inflammation. 

### 1.1. NLRP3

#### 1.1.1. The NLRP3 Inflammasome—Structure and Activation

NLRP3 belongs to the group of inflammasome-forming NLR proteins. Formation of the NLRP3 inflammasome occurs via oligomerization of NLRP3 monomers to NLRP3 oligomers, which leads to the recruitment of ASC via the PYD domains of NLRP3 and ASC. This in turn leads to the recruitment of pro-caspase-1, which via its CARD domain associates with the CARD domain of ASC [116]. The NLRP3-ASC-pro-caspase-1 complex facilitates the autocatalytic activation of pro-caspase-1, which in turn cleaves pro-IL-1β and pro-IL-18, but also Gasdermin D (GSDMD), into their active forms. GSDMD forms pores in the cell membrane for the release of IL-1β and IL-18 and initiates pyroptosis, a pro-inflammatory form of cell death [117,118,119,120] (Figure 2, panel 1). Given that IL-1β is a highly potent pro-inflammatory cytokine, NLRP3 inflammasome activation is tightly controlled, requiring a two-step activation process in most cells. The first, called priming, is mediated by the activation of TLRs, NLRs or cytokine receptors leading to the activation of NF-κB. NF-κB in turn induces the transcription of *NLRP3*, which at baseline is thought to be insufficiently expressed in the cell for inflammasome activation, and *pro-IL1b*. *ASC*, *pro-caspase-1* and *pro-IL18*, in contrast, are not upregulated by the priming signal [121,122]. The second, activating signal has been shown to be mediated by ionic fluxes, most prominently potassium efflux [51,52,123,124,125,126], but also changes in calcium concentration [127], sodium influx [51,128] or chloride efflux [129,130], as well as mitochondrial ROS, oxidized mitochondrial DNA [131,132,133] and lysosomal damage due to crystalline substances [19,134,135,136,137,138]. Given the biological diversity of these signals, it is highly unlikely that NLRP3 recognizes them directly. Instead, NLRP3 was proposed to react to a common upstream cellular signal that so far, however, has not been identified. IL-1β and IL-18 are key inflammatory mediators and their association with obesity and obesity-associated diseases, such as T2DM, atherosclerosis and NAFLD, is outlined below. 

#### 1.1.2. The NLRP3 Inflammasome in Adipose Tissue Inflammation, Insulin Resistance and T2DM

Compared to normal weight controls, both obese humans [58,78,79,80,81,139] and obese mice [58,59,60] present with increased expression of NLRP3 inflammasome components and IL-1β on mRNA and protein levels in the AT. The increase in expression is hereby mainly driven by ATMs [78,79,80] and correlates with the occurrence of IR [59,79]. Conversely, weight loss in obese human subjects normalized levels of *NLRP3* and *IL1b*, which coincided with improved insulin sensitivity [59]. Studies in mice show that deletion of *Nlrp3* protects against HFD-induced obesity, IR, dyslipidemia as well as infiltration of macrophages into the AT [59,63,64,140], and leads to improved serum glucose and insulin levels and insulin signaling in liver and cardiac tissue [64,65,140]. In human visceral adipocytes, *NLRP3* silencing reduces AT fibrosis [139]. Additionally, it was shown that increased caspase-1 levels after HFD feeding regulate insulin sensitivity and adipocyte differentiation. Concomitantly, pre-adipocytes from *caspase-1*- or *Nlrp3*-deficient mice present with higher insulin sensitivity and this effect was shown to be mediated by caspase-1-induced IL-1β processing [63]. Moreover, a positive association between the NLRP3 inflammasome and priming of murine ATMs toward a pro-inflammatory (M1) phenotype has been established [59], a process that contributes significantly to chronic low-grade inflammation in obesity [141]. In contrast, two other studies did not find protection of *Nlrp3^−/−^* or *caspase-1^−/−^* mice from HFD-induced weight gain, AT inflammation and elevated blood glucose levels [142,143], which might be explained by differences in diet composition, genetic background or husbandry-associated factors. Collectively, most studies show the NLRP3 inflammasome and IL-1β to play detrimental roles in AT inflammation and especially blood glucose regulation. As *Nlrp3* KO in mice in most studies leads to improved AT inflammation and glycemia, NLRP3 inhibition in humans appears as a promising treatment option for obesity-induced IR and T2DM. Indeed, treatment of T2DM patients with the human recombinant IL-1R antagonist anakinra has been shown to lower plasma glucose levels and to improve insulin secretion [85]. The pronounced effect of excess NLRP3 inflammasome activation and IL-1β release on the development of IR and its progression to a manifest T2DM is due to IL-1β interfering with the insulin signaling. It was shown that prolonged or chronic IL-1β treatment reduced glucose transporter 4 (GLUT4) expression and insulin-mediated Glut4 membrane translocation and glucose uptake in 3T3-L1 adipocytes (Figure 2, Panel 1). In addition, insulin receptor substrate (IRS)-1 expression and tyrosine phosphorylation, which mediates the docking of IRS-1 to the insulin receptor and thus enables signaling, was downregulated in IL-1β-treated 3T3-L1 cells and human adipocytes [62,144] (Figure 2, panel 1). The interference of IL-1β with glucose metabolism is further supported by a study finding IL-1 receptor (IL-1R)-deficient mice to be protected from HFD-induced AT inflammation and impaired glucose sensitivity [145] and by another study proving NLRP3 inflammasome activation in pancreatic islet-infiltrating macrophages to lead to β cell dysfunction [146]. Moreover, the *NLRP3* rs10754558 polymorphism, which leads to higher production of IL-1β [147] and thus resembles a gain-of-function mutation, and the *IL1B* rs16944 polymorphism have been associated independently by various groups with T2DM [148,149,150] and obesity [150]. Thus, the detrimental effect of the NLRP3 inflammasome on glycemia and insulin signaling demonstrated mainly in animal studies seems to be primarily mediated by IL-1β and its downstream signaling events, which highlights inhibition of NLRP3 or IL-1β as a potential therapeutic target in humans. 

NLRP3 activation in IR and T2DM has been shown to be mostly mediated by two different triggers, islet amyloid polypeptide (IAPP) and free fatty acids (FFAs). IR causes the pancreas to secrete more insulin, which also increases the secretion of IAPP. IAPP has a tendency to form aggregates that can cause lysosomal rupture in pancreatic macrophages and thus activate the NLRP3 inflammasome [151], leading to IL-1β-mediated destruction of the pancreatic β cells [152] and thus to a lack of insulin production (Figure 2, panel 4). Consistently, *Nlrp3*-deficient mice on HFD are protected from β cell fibrosis [64] (for recent reviews on this, see [153,154]). In addition to IAPP aggregates, FFAs are considered potential activators of the NLRP3 inflammasome in the context of IR and T2DM. Saturated fatty acids (SFAs) have been shown in vitro to activate the NLRP3 inflammasome. Stimulation of activated primary human monocytes [86] and murine bone marrow-derived macrophages (BMDM) [59,65,155] with palmitate or ceramides results in an NLRP3-dependent increase in IL-1β secretion. That ceramides also physiologically contribute to NLRP3 activation and development of IR and T2DM has been confirmed in diet-induced obese mice [59]. Interestingly, *Nlrp3*-deficient mice present with reduced serum levels of ceramides and LPS, which was proposed to be due to changes in gut microbiome composition [156]. For a detailed overview of the molecular details, we would like to refer the reader to the review article by Legrand-Poels et al. [157]. Thus, lowering the levels of circulating FFAs could provide an efficient strategy to prevent excessive NLRP3 activation in obese individuals.

In the context of obesity, the priming signals that drive the expression of *NLRP3* and *pro-IL1b* in the AT are not yet clearly identified. Earlier, it was proposed that FFAs, whose levels are increased in obesity, can activate NF-κB via TLR4 in macrophages and adipocytes [43] and thus lead to NLRP3 inflammasome priming. In line, adipocytes that are co-cultured with macrophages derived from HFD-fed mice present with elevated *Nlrp3* expression [158], and SFA-rich HFD feeding can increase the expression of *Nlrp3* and *Il1b* in the AT of mice and prime the Nlrp3 inflammasome [61]. Additionally, palmitic acid via dimerization of TLR1/2 can increase the expression of *pro-IL1b* in THP-1 cells and primary human monocytes [87]. However, a recent report by Lancaster et al. debunked the ability of SFAs to directly activate TLR4 but showed that TLR4 is involved in fatty acid-induced pro-inflammatory signaling [159]. Thus, how exactly SFAs prime the NLRP3 inflammasome remains to be elucidated. Oxidative stress and the resulting generation of ROS have also been proposed as the NLRP3 priming signal in the context of obesity. Excess nutrient intake leads to mitochondrial dysfunction and ROS production [160,161,162]. ROS have been shown to mediate NLRP3 priming as treatment of immortalized macrophages with a ROS inhibitor dose-dependently reduced *Nlrp3* expression [163]. Interestingly, ROS have also been implicated in inflammasome activation [131,164]. Additionally, hypoxia, which is induced in the AT of obese individuals [12], leads to increased ROS production [165] and can mediate NF-κB activation and NLRP3 priming [166]. Besides ROS production, hypoxia potentially triggers hypertrophic adipocyte death [12], leading to the release of DAMPs in the extracellular milieu, which in turn can induce NF-κB activation and inflammasome priming. Additionally, obese adipocytes have been shown to undergo NLRP3-dependent caspase-1-triggered pyroptosis [167], which again leads to the release of DAMPs (Figure 2, panel 1). Concomitantly, *NLRP3* has been shown to be upregulated in human visceral adipocytes under hypoxic conditions [139]. Besides fatty acids, ROS and AT-derived DAMPs, hyperglycemia [168] and endoplasmic reticulum (ER) stress in AT and pancreas [169,170] have been associated with NLRP3 inflammasome priming. Along these lines, advanced glycation end products (AGEs), resulting from non-enzymatic glycation of plasma proteins during hyperglycemia and thus increased in T2DM patients [171], have been shown to upregulate *NLRP3* expression [172]. When given to mice, AGEs induced β cell apoptosis and reduced glucose tolerance, which was ameliorated by *Nlrp3* deficiency [66]. Additionally, serum amyloid A (SAA), a liver-derived acute phase protein whose levels are correlated with T2DM [173], increases the expression of *NLRP3* in macrophages [174]. Lastly, the short-chain fatty acid sensors G protein-coupled receptor 43 (GPR43) and GPR109a might be involved in NLRP3 signaling [175]. GPR43 seems to be implicated in priming the NLRP3 inflammasome, as *Gpr43*-deficient mice present with lower inflammasome activation and IL-1β processing [176]. The latter might be of relevance upon high gut permeability and the resulting increased serum levels of butyrate and acetate, which might also occur in postprandial conditions. 

In summary, NLRP3 is a key factor for low-grade AT inflammation and IR in obese individuals. The NLRP3 inflammasome thereby is primed and/or activated by several metabolites such as FFAs, IAPP aggregates, AGEs and SAA and their downstream effects including ROS induction, release of DAMPs, hyperglycemia, ER stress and GPR signaling. Inhibiting one or combined inhibition of several of these processes and factors thus could serve as therapeutic option for treating AT inflammation and disturbed glycemia.

#### 1.1.3. The NLRP3 Inflammasome in Atherosclerosis

In addition to its role in IR and AT inflammation, activation of the NLRP3 inflammasome has also been implicated in the development and progression of atherosclerosis, a chronic inflammatory disease of the arterial walls [19]. In the course of the disease, atherosclerotic plaques build up in the vascular endothelium and cause narrowing of the arterial lumen. These plaques consist of lipid depositions, especially low-density lipoprotein (LDL) cholesterol, macrophages and other infiltrating immune cells, which mediate the pro-inflammatory signaling [177]. Patients with coronary atherosclerosis present with increased mRNA and protein levels of NLRP3, ASC, caspase-1, IL-1β and IL-18 in atherosclerotic plaques, the amount of NLRP3 correlating with disease severity [82,83,84]. Based on studies with atherosclerosis-prone apolipoprotein E (*apoE*) or LDL receptor (*Ldlr*)-deficient mice fed a high fat/high cholesterol diet, it has been shown that deficiency of *Nlrp3*, *caspase-1*, *Asc*, *Il1b* or *Il18*, as well as blockade of IL-18, results in a reduction in number and size of atherosclerotic lesions and a higher stability of atherosclerotic plaques [19,68,69,70,71,73]. Bone marrow (BM) transplantation transferring BM from *caspase-1/11-*, *Nlrp3-*, *Asc-*, *Il1b-* or *Il18*-deficient mice to *Ldlr^−/−^* animals, showed that inflammasome-mediated inflammatory signaling in the hematopoietic compartment, especially in macrophages, is critical for atherosclerosis development [19,68]. Additionally, in *Ldlr*-deficient mice, an Nlrp3-dependent reprogramming of myeloid precursors, reflected by increased activation potential of granulocytic and monocytic progenitor cells, after WD feeding was observed, and this pro-inflammatory phenotype was preserved even after switching to a normal diet (chow diet, CD) for four weeks. Also in this model, *Nlrp3* deficiency results in significantly smaller atherosclerotic lesions compared to WT animals [72], highlighting the importance of both the NLRP3 inflammasome in the pathogenesis of atherosclerosis and of the dietary intake as subordinate activation signal. However, one study did not find beneficial effects of *Nlrp3*, *Asc* or *caspase-1* knockout (KO) in *Apoe*-deficient mice on HFD [178]. These discrepancies might be explained by differences in the mouse models used but also might point to an NLRP3-independent progression of atherosclerosis, eventually driven by the inflammasome-independent cytokine IL-1α. In fact, two studies using *Il1a* and *Il1b* KO mice have demonstrated a more pronounced effect for *Il1a* compared to *Il1b* deficiency on the reduction in atherosclerotic lesions [179,180]. Nevertheless, most studies provide clear evidence identifying the NLRP3 inflammasome and IL-1β as major drivers of atherosclerosis development and progression. As deficiency in components of the Nlrp3 inflammasome in the mouse model protects against atherosclerosis development, inhibition of the NLRP3 inflammasome also in humans appears as a promising treatment option. In fact, clinical data from the CANTOS trial demonstrated lower incidence of myocardial infarction, non-fatal stroke and cardiovascular death in cardiovascular risk patients receiving the monoclonal antibody canakinumab targeting IL-1β [181]. 

In contrast to IR and AT inflammation, the activational triggers of the NLRP3 inflammasome in atherosclerosis are well defined and crystallized oxidated LDL (oxLDL) has been identified as one of the main activators in atherosclerotic lesions. Cholesterol, an essential lipid in vertebrates, is insoluble in aqueous solutions [182] and especially LDL cholesterol is susceptible to ROS-mediated oxidation [183]. Cholesterol crystals are considered hallmarks of atherosclerotic lesions [184] and are taken up by macrophages and cause lysosomal rupture, resulting in the release of cholesterol crystals and other lysosomal content, such as cathepsin B and L, in the cytosol, which in turn leads to the induction of caspase-1 cleavage and IL-1β secretion [19,185] (Figure 2, panel 3). Endocytosis of oxLDL and subsequent crystallization have been shown to be mediated by the PRR cluster of differentiation 36 (CD36), as *Cd36*-deficient BMDMs failed to secrete IL-1β upon oxLDL stimulation and *Apoe^−/−^ Cd36^−/−^* mice on WD presented with smaller atherosclerotic lesions and reduced IL-1β serum levels compared to *Apoe^−/−^* mice [186]. As it has been shown that minimally modified LDL can prime cells for NLRP3 inflammasome activation [19], LDL in atherosclerotic plaques seems to be sufficient to mediate both the priming and the activation of the NLRP3 inflammasome. Intake of oxLDL crystals and the resulting lysosomal rupture and inflammasome activation cause pyroptosis and necroptosis of macrophages [187], leading to the release of intracellular molecules to the extracellular milieu, where they are recognized as DAMPs. Extracellular ATP, for example, has been shown to activate the NLRP3 inflammasome in atherosclerotic plaques via the purinergic receptor P2X_7_ whose activation allows the influx of calcium and the efflux of potassium [188,189], known triggers for inflammasome activation (see above) (Figure 2, panel 3). P2X_7_ expression was increased in human [189] and murine [188] atherosclerotic lesions and deficiency in *p2x_7_* led to reduced caspase-1 and inflammasome activation upon stimulation with LPS and oxLDL, respectively, and reduced atherosclerotic lesions and macrophage recruitment in murine models of atherosclerosis [188,189]. Interestingly, also independent of atherosclerosis, cholesterol metabolism and the NLRP3 inflammasome appear to be functionally related. High cholesterol biosynthesis causes strong activation of the NLRP3 inflammasome via translocation of the transcription factor complex sterol regulatory element-binding protein (SREBP) cleavage-activating protein (SCAP)-SREBP2 from the ER to the Golgi apparatus, which may contribute to chronic, low-threshold inflammation in obese individuals [190]. 

In summary, NLRP3 and IL-1β play an important role in the development and progression of atherosclerosis and in this context are primarily activated by cholesterol crystals and extracellular ATP. 

#### 1.1.4. The NLRP3 Inflammasome in NAFLD and NASH

While the evidence on a contribution of NLRP3 to glucose metabolism, AT inflammation and atherosclerosis is clear, in the context of NAFLD, which in 20% of the cases progresses to a condition of chronic liver inflammation (non-alcoholic steatohepatitis, NASH) [191,192,193], the data are less consistent. On the one hand, and in contrast to the data described above, *Nlrp3-*, *Nlrp6-*, *Asc-* and *caspase-1*-deficient mice present with a more severe course of NAFLD/NASH than WT controls [76] (Figure 2, panel 5). NLRP6, like NLRP3, is able to form an inflammasome and has been implicated in the modulation of the gut microbiome and in microbial host defense [194]. This protective effect of the NLRP3 and the NLRP6 inflammasomes in NAFLD/NASH has been shown to be mediated by IL-18, whose deficiency can result in changes of the microbiota composition and increased translocation of TLR4 and TLR9 agonists to the portal circulation, exacerbating hepatic steatosis and inflammation in mice fed a methionine–choline-deficient diet (MCDD) (Figure 2, panels 2 and 5) [76]. In line, co-housing with *Il18-* or *Asc*-deficient mice led to increased steatosis and obesity in WT animals and antibiotic treatment reduced disease severity in *Asc*-deficient mice [76]. These data highlight the importance of inflammasomes not only as crucial regulators of inflammatory processes, but also as modulators of the gut microbiome, where *Nlrp3* and especially *Nlrp6* deficiency have been shown to alter the microbial composition and to increase colitis-susceptibility and inflammation-associated tumorigenesis [76,195,196]. However, despite the seemingly beneficial influence of NLRP3 and NLRP6 on the stability of the intestinal barrier and subsequently liver homeostasis, a protective effect of *Nlrp3* deficiency in NAFLD and liver inflammation and fibrosis in mice on a choline-deficient amino acid-defined (CDAA) diet was shown, which was attributed, among other things, to the lower production of IL-1β [67] (Figure 2, panel 5). Additionally, two other studies reported protective effects of *Nlrp3* or *Asc* deficiency in the development of HFD-induced liver steatosis [59,140] and hepatic insulin sensitivity [140]. Consistently, inflammasome components were upregulated in samples from NASH compared to NAFLD patients or healthy volunteers [67,139] and in the liver of obese T2DM patients compared to normal-weight individuals [139]. These results contradict the abovementioned protective role of ASC inflammasomes in NAFLD; the discrepancy could be explained by the different feeding duration of the NAFLD-inducing diets [59,67,76,140]. It is possible that in the initial phase of NAFLD, NLRP3 and NLRP6 act by inducing IL-18 secretion and thus its beneficial effects on the microbiota to protect and slow the progression of the disease, whereas with prolonged persistence of NAFLD-inducing triggers, the adverse, pro-inflammatory effect of IL-1β predominates and NLRP3 then contributes to the progression of liver degeneration, inflammation, and fibrosis. Based on this theory, a timing-dependent rather than a general ASC inflammasome inhibition appear as a potential therapeutic option, with beneficial effects being achieved by inhibition of the inflammasome only when the liver inflammation has already been established. 

#### 1.1.5. The NLRP3 Inflammasome in Physiological Inflammation

Most of the work discussed above depicts the NLRP3 inflammasome as a detrimental factor that enhances a pro-inflammatory phenotype in obesity and whose absence or inhibition seems to be beneficial. However, inhibition of the IL-1 signaling pathway and inflammasome formation also has been reported to have negative effects on body weight and glucose homeostasis. Deficiency in the *Il1r* and double KO of *Il1* and *Il6* in mice on a normal (low-fat) diet leads to the spontaneous development of obesity, manifesting in higher body weight gain, IR and decreased responsiveness to the adipokine and satiety hormone leptin [74,75]. Obesity-prone *db*/*db* mice lacking *Asc* on HFD also gain more weight and present with impaired glucose homeostasis [76]. Conversely, mice deficient in the IL-1 receptor antagonist (*Il1ra^−/−^*), known to limit IL-1 signaling, remain lean and present with a defect in lipid accumulation and decreased insulin, leptin and triglyceride levels, even under HFD feeding [75,197]. These results are in clear contrast to the work discussed above, in which deficiencies in the IL-1R, in IL-1β or in components of the inflammasome protect against obesity-associated morbidities. Key to these differences and the main technical difference between the studies showing beneficial and the ones proving detrimental effects of the NLRP3 inflammasome on body weight and glucose homeostasis is the diet administered to the mice. In the obesity context, where the NLRP3 inflammasome confers detrimental effects, HFD and WD were used in almost all studies. In contrast, studies that support a beneficial role of the NLRP3 inflammasome to protect against weight gain largely used normal chow/low-fat diets. This opens the possibility that IL-1β plays a dual role in metabolism and inflammation. NLRP3 and IL-1β appear to contribute to the maintenance of normal body weight and metabolic homeostasis on an isocaloric diet, whereas the adverse, pro-inflammatory effects of IL-1β predominate on a high-calorie diet. In line, mice become hypoglycemic after injection of sub-inflammatory doses of IL-1β and present with increased insulin levels but respond normally to glucose challenge, indicative of no impairment of the glycemic control [198]. By contrast, sustained elevation of IL-1β is associated with T2DM, leading to β cell dysfunction and cell death via activation of macrophages in the pancreas [45]. The hypothesis of a dual function for IL-1β in glucose metabolism is further supported by a more recent study showing that postprandial increases in blood glucose levels in mice leads to increased macrophage-derived IL-1β production, which in turn enhances insulin secretion, a process that depends on IL-1R expression on pancreatic β cells. Insulin in turn increases IL-1β secretion of macrophages and both, insulin and IL-1β, stimulated glucose uptake, with IL-1β preferably stimulating glucose uptake by immune cells. In line, *Il1b* KO or macrophage depletion leads to reduced insulin secretion and increased blood glucose levels, and injection of IL-1β potentiated glucose-stimulated insulin secretion [34]. Additionally, some of the protective effects of the NLRP3 inflammasome may also be mediated via IL-18, as deficiency of *Il18* in diabetes-prone non-obese diabetic mice induces the spontaneous development of obesity, hyperglycemia and IR, which can be reversed by IL-18 administration [77]. Thus, in metabolically healthy individuals, the intact NLRP3 inflammasome and IL-1 signaling seem to be needed for body weight and blood glucose homeostasis, and especially blood glucose homeostasis depends on postprandial nutrient-induced, low-level IL-1β-mediated inflammation. Transferring the data generated mainly in mice to humans would mean that inhibition of the NLRP3 inflammasome or IL-1β, as promising as it appears for treating obesity-associated morbidities, is not a suitable option for prevention of adiposity and its related metabolic diseases.

Altogether, a large body of evidence demonstrates that the NLRP3 inflammasome contributes significantly to the development and progression of obesity-associated comorbidities such as IR, T2DM, atherosclerosis and NAFLD. This is mediated largely by IL-1β and, in some diseases (atherosclerosis), additionally by IL-18. However, the role of the NLRP3 inflammasome in metabolism is more complex and not always negative. Upon leveled energy balance, the NLRP3 inflammasome appears to maintain metabolic homeostasis by ensuring normal body weight and glucose metabolism through a physiological inflammation mediated by IL-1β and IL-18. In addition, in early stages of NAFLD, IL-18 may partially compensate for the negative effects of IL-1β and slows disease progression. Why and/or at which point these protective effects of the NLRP3 inflammasome are lost and the adverse mechanisms become dominant should be the subject of future research. In any case, the NLRP3 inflammasome represents an interesting starting point for the therapy of low-grade inflammation in obesity and the availability of clinical antagonists of IL-1β opens new possibilities for the therapy of T2DM and cardiovascular diseases in the context of obesity [199].

### 1.2. NOD1 and NOD2

NOD1 and NOD2 are cytosolic PRRs that sense bacterial PGN fragmentsto initiate pro-inflammatory responses [200]. The minimal ligand for NOD1 is γ-D-glutamyl-meso-diaminopimelic acid (iE-DAP), which is part of the cell wall of Gram-negative and certain Gram-positive bacteria [201,202], while NOD2 recognizes muramyl dipeptide (MDP), which is present in both bacterial categories [203,204]. Upon activation, NOD1 and NOD2 homodimerize and recruit RIPK2 (RIP2), which initiates inflammation via NF-κB activation and the induction of adaptive immune responses to defend against intracellular bacteria. Additionally to these classical pro-inflammatory functions, NOD1 and NOD2 signaling in the gut contributes to protecting the intestinal epithelium from bacterial invasion and to maintaining gut homeostasis by promoting the secretion of antimicrobial peptides [205].

Beside their functions as PRRs in antibacterial immune defense, NOD1 and NOD2 more recently have been reported to contribute to metabolic diseases, especially in the context of IR. While NOD2 mainly has been shown to protect against T2DM, NOD1 promotes the development of IR and blood glucose dysregulation. Correlation studies found an association between *NOD1* and *NOD2* expression and glucose intolerance. In T2DM patients, *NOD1* and *NOD2* expression is increased in CD14^+^ monocytes, and correlates with disease progression, TNF-α and IL-6 serum levels and the abundance of pro-inflammatory markers such as CD11b and CD36 on monocytes [97]. This is in line with two studies showing *NOD1* expression to be enhanced in omental and subcutaneous AT of women with gestational diabetes compared to healthy pregnant women [96] and in the abdominal subcutaneous AT of MetS patients [98]. In the latter study, NOD1 protein levels additionally correlated with metabolic parameters such as waist circumference, HbA1c, IR and circulating pro-inflammatory cytokines [98]. Additionally, in differentiated human adipocytes in vitro, *NOD1* and *NOD2* levels were upregulated concomitantly with increased NF-κB translocation and pro-inflammatory cytokine secretion. Interestingly, although both NLR proteins signal via the same downstream pathway, only NOD1 but not NOD2 stimulation results in NF-κB activation, impaired insulin signaling and reduced GLUT4-dependent glucose uptake [93,96,98,206]. This was also observed in murine hepatocytes, which express both Nod1 and Nod2, but only activation of Nod1 resulted in NF-κB-dependent cytokine release [207]. In line, Zhang et al. showed that HFD-induced ER stress in mice results in upregulated *Nod1* expression, which then promotes pro-inflammatory signaling and low-grade inflammation via Ripk2. Even though *Nod2* was expressed and upregulated as well, on a functional level only Nod1 stimulation induced the pro-inflammatory response [208]. Thus, although NOD1 and NOD2 are functionally closely related and both can activate RIPK2-dependent signaling, their effects on glucose metabolism differ, which might be partly due the presence of a second CARD domain in NOD2 that is absent in NOD1. Based on these differences, NOD1 inhibition might serve as a therapeutic option to alleviate obesity-induced glucose intolerance, while NOD2 stimulation might favor blood glucose homeostasis.

### 1.3. NOD1

In vivo experiments confirmed the detrimental role of NOD1 in obesity, as *Nod1* and *Nod2* double KO mice showed reduced HFD-induced IR, lipid accumulation, adipocyte size and inflammation in liver and AT compared to WT mice on HFD. Furthermore, Nod1 but not Nod2 stimulation increased peripheral and hepatic insulin intolerance and serum levels of pro-inflammatory cytokines in WT mice [88] (Figure 2, panels 1 and 5). Concomitantly, in another study, *Nod1^−/−^* mice on HFD presented with less pro-inflammatory cells in liver and AT. Additionally, reduced resting energy expenditure and a higher body weight was observed in these animals [91]. This effect of NOD1 on IR and fat mass might need some time to establish, as one study could not observe this phenotype after a 4-week HFD [209]. The other studies in comparison used a feeding duration of 16 [88] and 6 [91] weeks. *Nod1* KO in hematopoietic cells is sufficient to abolish the HFD-induced AT inflammation and IR but does not prevent body weight gain. Mechanistically, due to the lack of *Nod1* in immune cells, there are less pro-inflammatory macrophages in the AT, resulting in lower secretion of the chemokine CXCL1 and subsequent neutrophil recruitment [7]. This highlights the *Nod1* expression in immune cells to be a major driver of the abovementioned metabolic changes (Figure 2, panel 1). The influence of NOD1 activity on low-grade inflammation and IR was also demonstrated by injections of NOD1 ligands into WT mice, which increased peripheral and hepatic IR and serum levels of pro-inflammatory cytokines such as CXCL1 [88]. Ripk2 signaling contributes to this process, as blood glucose levels did not change in *Ripk2^−/−^* mice after NOD1 ligand injection [210]. The physiological relevance of NOD1 activators in the serum has been proven by studies showing a time-dependent increase in circulating NOD1-stimulating PGN fragments after HFD (Figure 2, panel 1) that was associated with impaired insulin signaling and glucose tolerance [7,90]. Thus, inhibition of RIPK2 downstream from NOD1 or the reduction of circulating NOD1 agonists after meal intake appear as possible intervention strategies to mitigate the detrimental effects of NOD1 on glucose metabolism. Additionally, *Nod1* has been shown to be upregulated by HFD in mouse macrophages, skeletal muscle, AT and liver [90]. One mouse study, however, reported a rather contradictory role for NOD1 in IR, as the injection of gut bacterial extract improved glucose tolerance, but not in *Nod1^−/−^* mice, indicating that under certain circumstances NOD1 signaling might be required for glucose homeostasis [211]. Further research is needed to elucidate the factors determining the nature of NOD1′s influence on glycemia in order to develop a reliable treatment strategy involving NOD1 signaling regulation. On a molecular level, serum NOD1 ligands can prime neutrophils in the bone marrow, which then are able to defend more effectively against bacterial pathogens [212], a mechanism that in metabolically healthy individuals eventually helps a more efficient defense against invading pathogens but in metabolically unhealthy individuals might contribute to metabolic dysbalance. Upon HFD, in addition to PGN fragments, low amounts of bacteria can also translocate from the intestine into the blood and AT and stimulate pro-inflammatory cytokine expression [209]. Additionally, HFD promotes adherence of *Escherichia coli* to the ileum mucosa and their co-localization with dendritic cells [209], promoting inflammation in the gut. Although *Nod1^−/−^* mice do not present with dysbiosis in major bacterial groups, the deficiency in *Nod1* weakens the epithelial barrier function due to reduced expression of Nod2, mucin-2 (Muc2), defensins and keratinocyte-derived chemokine (KC) [213]. This suggests that NOD1, in addition to inducing inflammation upon recognition of gut-derived PGN, might regulate intestinal barrier stability and therefore prevent translocation of bacteria or bacterial metabolites into the blood, in turn preventing inflammation and IR. As most of these studies were performed in mice and comparable experiments with humans are difficult, it might be interesting to measure NOD1-activating PGN fragments in the blood of obese people and after HFD intake.

Furthermore, diverse effects of NOD1 in white AT and adipocytes have been described. Murine and human pre-adipocytes constitutively express *Nod1* but not *Nod2*. Interestingly, stimulation with different MAMPs increases *Nod2*, but not *Nod1* expression and activation of Nod1 in these cells augments NF-κB-dependent IL-6 secretion [214]. *Il6* expression is upregulated during adipogenesis [215], but its effect on AT seems to be depot-specific. In visceral AT, IL-6 type signaling increases FFA secretion, which promotes hepatic IR and steatosis. On the other hand, IL-6 enhances the secretion of leptin from subcutaneous AT, which promotes glucagon-like peptide 1 (GLP-1) release and subsequently benefits insulin signaling [216]. It was also reported that NOD1-activating PGN fragments inhibit adipocyte differentiation of human adipose-derived adult stem cells [93] and murine 3T3-L1 cells [93]. Nod1 stimulation also increased lipolysis in white AT of WT but not *Nod1^−/−^* mice and in 3T3-L1 adipocytes [89,94] (Figure 2, panel 1). However, the signaling pathway behind this process is still controversially discussed. In 3T3-L1 cells, the increased lipolysis leads to accumulation of diacylglycerol (DAG) and induction of a cell autonomous inflammation via protein kinase C (PKC) δ with subsequent interleukin-1 receptor-associated kinase (IRAK) 1/4 activation, which results in enhanced secretion of the pro-inflammatory cytokines IL-1β, IL-18, IL-6, TNF-α and monocyte chemoattractant protein 1 (MCP-1) [94]. Other studies using mouse models reported that Nod1-mediated lipolysis requires protein kinase A (PKA), NF-κB and hormone-sensitive lipase (HSL) [89,95] or Ripk2 [210]. In addition to inducing lipolysis, Nod1 activation impairs insulin signaling and glucose uptake by increasing the release of pro-inflammatory cytokines in 3T3-L1 adipocytes [88,92]. Thus, in white AT, NOD1 inhibits differentiation of adipocytes and induces lipolysis and pro-inflammatory cytokine secretion. As selective targeting of NOD1 expression in specific cell types in humans is not feasible, understanding the signaling pathway behind these effects will help establish the way toward development of novel therapeutic approaches.

Nod1 activation also impairs differentiation of brown AT cells. The expression of the key differentiation factor, peroxisome proliferator-activated receptor (PPAR) γ, and lipid accumulation was reduced in these cells upon NOD1 activation [217]. In brown fat tissue of obese mice, Nod1, Nod2 and pro-inflammatory cytokine mRNA is expressed, but during differentiation only *Nod1* and mRNA expression of uncoupling protein 1 (UCP-1) is upregulated. After activation of Nod1 in a brown fat cell line, Ucp-1 expression and oxygen consumption was decreased [218]. A reduction in Ucp-1 expression was also observed in *Nod1^−/−^* mice fed a HFD [91] and after chronic activation of Nod1 in undifferentiated brown AT cells [217]. So, in brown AT, in addition to the inhibition of differentiation, NOD1 activation reduces energy expenditure by decreasing UCP-1 expression, highlighting a possible contribution of NOD1 to obesity not only in white but also in brown AT. Increasing UCP-1 expression here might serve as a possible starting point to counteract NOD1-mediated reduced energy expenditure. 

Other than the effects of Nod1 on adipocytes, some studies suggest a more direct role for NOD1 in fatty acid sensing [219,220]. FFAs from macronutrient intake and gut microbial metabolism could therefore play an important role in the activation of NLRs, as discussed above for NLRP3. In adipocytes and human colonic epithelial cells (HCT116), an oleate/palmitate mix or lauric acid induced NOD1-dependent NF-κB activation and pro-inflammatory cytokine expression [219,220]. Additionally, insulin-stimulated glucose uptake in adipocytes was impaired after fatty acid treatment [220]. Interestingly, the pro-inflammatory reaction was reduced after stimulation with n-3 polyunsaturated fatty acids [219]. These findings are supported by a report where the Lys/Lys variant of the Glu266Lys polymorphism in *NOD1* in combination with a high intake of SFA was correlated with increased risk for diabetes [221]. Thus, in addition to circulating PGN fragments, NOD1 activation is likely also mediated by FFAs, which might play a role especially in the AT. How fatty acids can activate NOD1 at the molecular level remains elusive at present. Clarifying the molecular mechanism of fatty acid sensing might help to understand and mitigate induction inflammation due to food intake in obese individuals.

### 1.4. NOD2

Although closely related to NOD1, there is increasing evidence that NOD2 signaling, in contrast to NOD1, protects from low-grade inflammation and IR. *Nod2^−/−^* mice show increased body weight gain, visceral AT and fat accumulation in the liver, as well as augmented macrophage infiltration in the AT and a reduced glucose tolerance, and these effects are observed under normal diet [99] and HFD [100,101] (Figure 2, panel 5). Additionally, the T helper cell (Th) 17/Th1 balance is disturbed in favor of Th1 responses in secondary lymphatic organs [101]. In line, in leptin-deficient *ob*/*ob* mice and DIO mice stimulation of Nod2 has no effect on body weight but reduces AT inflammation and hepatic IR and results in lowered expression of various pro-inflammatory molecules in the white AT [103]. Mechanistically, it was proposed that this is mediated by NOD2-specific activation of interferon regulatory factor 4 (Irf4) [103]. Thus, an anti-inflammatory role in metabolism can be attributed to NOD2 and NOD2 stimulation in obese could serve as strategy to reduce AT inflammation and IR. In contrast to Nod2, Nod1 stimulation impairs glucose tolerance, which is independent of Irf4, indicating activation of different signaling pathways by Nod1 and Nod2, leading to the different outcomes [103]. Additionally, in vivo, adipocyte Irf4 and Ripk2 expression was proven necessary for MDP to decrease blood glucose in low endotoxemia during HFD, but *Irf4* deficiency in this study did not impair the reduction in pro-inflammatory genes induced by Nod2 activation and in contrast to the function of Irf4 in myeloid cells is male specific [222]. This indicates that pro-inflammatory cytokine secretion is independent of Irf4, whereas the NOD2-dependent effects protecting against IR require Irf4 signaling, thus highlighting Irf4 as a potential therapeutic target for decreasing IR. Furthermore, it was shown that Ripk2-dependent Nod2 stimulation in non-hematopoietic cells is critical for protection against IR and the reduction in low-grade inflammation [100,104]. Supporting the importance of NOD2 in non-immune cells, *Nod2* KO in the liver after HFD feeding results in augmented liver inflammation, lipid and cholesterol metabolism and increased collagen synthesis, which all promoted liver steatosis and fibrosis [102]. Additionally, in a hepatic tumor model, *Nod2* deficiency increased body weight, liver tumors, cell proliferation, cholesterol biosynthesis and the invasion of pro-inflammatory monocytes, T cells and neutrophils in mice [223]. A protective effect of NOD2 on IR is also supported by in vitro experiments revealing how NOD2 mechanistically contributes to improving insulin sensitivity. In human and murine L cells, Nod2 activation augments Glp-1 secretion, which downstream benefits insulin signaling. Interestingly, *Nod2* and *Glp1* are downregulated under hyperglycemic conditions [105]. In vivo, Nod2 stimulation increases Glp-1 serum levels, but does not improve glucose tolerance [105]. The effect of Irf4-mediated signaling, however, seems to depend on the tissue context as some in vitro experiments do not support the beneficial effect of NOD2 on IR. Nod2 stimulation, for example, decreased cell-autonomous insulin-stimulated glucose uptake in rat skeletal muscle cells [224] and impaired adipocyte differentiation in adipose-derived adult stem cells [93]. In RAW264.7 mouse macrophages, *Nod2* was upregulated after treatment with the adipokine resistin, which is associated with NF-κB activation and pro-inflammatory cytokine secretion, suggesting a pro-inflammatory role for Nod2 signaling in macrophages [225]. These discrepancies between in vitro and in vivo data on NOD2′s role in glucose homeostasis point toward a complex interplay between NOD2 and other factors and different cell types to mediate its beneficial effects on metabolism, which awaits further investigation in order to develop targeted therapeutic strategies involving NOD2 stimulation. 

The NOD2 agonist MDP is derived from the gut microbiota, and Nod2 expression can influence the composition of the gut microbiome in mice. Transferring microbiota of *Nod2^−/−^* to WT mice enhances the susceptibility of those animals to metabolic disorders [99], and there are data supporting that *Nod2* deficiency can augment the presence of bacteria in the normally non-colonized gut mucosa and bacterial translocation into the liver and AT on HFD [100]. Interestingly, hepatocyte-specific depletion of *Nod2* is also sufficient to induce changes in the gut microbiota [102]. Additionally, it was reported that full body depletion of *Nod2* reduces alpha diversity and leads to dysregulated tight junctions [101], whereas Nod2 stimulation in obese mice has no effect on microbiota composition [103]. This shows a reciprocal influence of NOD2, the microbiota and the metabolism in obesity.

In summary, NOD1 activation by gut microbiota-derived PGN fragments, which are present in the blood after HFD feeding, contributes to systemic low-grade inflammation and IR. This is largely dependent on NOD1 expression in hematopoietic cells but also in adipocytes. This suggests that a complex interplay between immune cells and adipocytes accounts for the NOD1-mediated effects in obesity and IR. For NOD2, different in vivo experiments suggest a protective role in low-grade inflammation and IR. NOD2 signaling, especially in non-hematopoietic cells such as hepatocytes, is important to protect against the metabolic changes. Disturbed NOD2 signaling can affect inflammation, fat accumulation and fibrosis in the liver. Additionally, NOD2 seems to influence the microbiota, although a direct effect of these NOD2-driven microbial changes on body weight is less clear. Still, little is known about the interplay between NOD1 and NOD2 and possible antagonistic effects, and the molecular determinants that are causal for the activation of NOD1 and NOD2 in obesity. Amid the successful development of specific pharmacological inhibitors of RIPK2 kinase [226,227], and synthetic NOD2 activators such as Murabutide [228] that can be used as postbiotics, clarification concerning the function of these proteins in obesity could open new possibilities for treating the MetS in the long term.

### 1.5. Other NLR Proteins

#### 1.5.1. NLRC5

NLRC5 shows the typical NLR architecture composed of a central nucleotide-binding domain (NBD), an N-terminal structurally atypical CARD domain, and C-terminal LRRs. However, the atypical long LRR (27 repeats) makes it the largest member of the NLR family [56,229,230,231,232]. NLRC5 is constitutively expressed in different tissues such as the spleen, lymph nodes, lungs and the intestinal tract [229], but highest levels are found in hematopoietic cells, especially lymphocytes [232]. In contrast to many other NLRs, no function of NLRC5 as a bona fide PRR has been described so far, and its activating ligand remains unknown. Instead, NLRC5 acts as the transcriptional master regulator of MHC class I genes [56]. MHC class I molecules are expressed on all nucleated cells and are responsible for cell-derived antigen presentation to CD8^+^ T cells to drive adaptive immunity [233]. So far, this is the primary function known for NLRC5. Additionally, NLRC5 has been described as a negative regulator for NF-kB [229,234] and type I IFN responses [230,234], and an activator of the NLRP3 inflammasome [235,236,237]. However, there are opposing results on immune response modulation by NLRC5, as the first reports on NLRC5 identified a positive role in type I IFN responses [231,232]. The role of NLRC5 in immune responses beyond MHC class I regulation thus remains controversial [238]. NLRC5 has also been implicated to participate in tumor progression, as one of the evasion mechanisms employed by malignant cells to escape CD8^+^ T cell-mediated immune responses is the downregulation of MHC class I expression. Concomitantly, it has been shown that NLRC5-expressing tumors are controlled more efficiently than control tumors [239,240]. 

Recent evidence now suggests a novel role for NLRC5 in metabolic traits. Two independent epigenome-wide association studies identified the *NLRC5* locus to be differentially methylated in normal weight versus obese individuals, but with conflicting results. Meeks et al. identified the methylation of the NLRC5 locus to be positively associated with BMI, obesity, and waist circumference in a Ghanaian cohort [109]. In contrast, Cao-Lei et al. found the *NLRC5* locus to be hypo-methylated in children with obesity compared to normal-weight children [110]. Moreover, *NLRC5* was identified as a candidate gene to affect HDL-Clevels in humans [111] and single nucleotide polymorphisms (SNP) in *NLRC5*, and its promotors have been associated with altered triglyceride levels and dyslipidemia [112,113]. In mice, *Nlrc5* deficiency was shown to alleviate HFD-induced diabetic nephropathy [106] but to aggravate myocardial damage [107]. Additionally, *Nlrc5* KO mice on HFD have been reported to gain more body weight [107]. This is in line with a very recent report where female *Nlrc5^−/−^* mice under HFD manifest increased weight gain and waist circumference, larger adipose tissues (epididymal and inguinal) and adipocyte size when compared to female WT mice [108]. In this study, NLRC5 in synergy with PPARγ, the key transcriptional regulator for the differentiation of adipocytes, enhanced the transcription of PPARγ target genes involved in lipid metabolism [108]. Functional interaction of NLRC5 and PPARγ was insentiently reported in human aortic smooth muscle cells [114]. Transferring these results to humans, genetic variants of NLRC5 leading to diminished NLRC5 functionality could favor the development of obesity. Additionally, a contribution of NLRC5 to PPARγ-mediated dampening of inflammatory responses upon LPS stimulation of BMDMs was shown [108]. Furthermore, NLRC5 has been proposed to play a role in liver fibrosis, which is an endpoint of NAFLD. However, conflicting results are reported [241,242,243,244], questioning the physiological relevance.

Together, these studies support novel roles of NLRC5 in metabolism and body weight regulation. The underlying mechanisms are still not fully understood but involve regulation of PPARγ and likely additional functions of NLRC5 such as negative regulation of inflammatory responses in low-grade inflammation.

#### 1.5.2. NLRP12

NLRP12 is another PYD containing NLR protein and is predominantly expressed in cells of myeloid-monocytic origin [245]. Summarizing the available literature, the view emerges that NLRP12 is a regulatory NLR protein, inhibiting canonical and non-canonical NF-κB and extracellular signal-regulated kinase (ERK) activation and negatively influencing the development of colitis and colon cancer [246,247]. However, in the context of bacterial infection, NLRP12 was reported to form an inflammasome and to contribute to the release of IL-1β and IL-18 and thus to inflammation [248]. Recently, NLRP12 was shown to protect against obesity by influencing the composition of the gut microbiota [115]. *Nlrp12*-deficient mice on HFD present with lower energy expenditure, higher body weight and greater body fat percentage and have increased insulin tolerance in the liver and white AT compared to WT mice [115]. Additionally, increased inflammasome activation and increased serum levels of TNF-α and IL-6 and numbers of pro-inflammatory M1-type macrophages in the AT were observed in *Nlrp12*-deficient animals. Myeloid-specific depletion of *Nlrp12* led to similar results compared to full body depletion, suggesting that loss of myeloid-specific *Nlrp12* confers this phenotype of obesity, IR and increased pro-inflammatory signaling. Raising full body *Nlrp12* KO mice in germ-free conditions prevented the higher body weight gain and reduced inflammation in these animals as did co-housing *Nlrp12* KO with WT animals during HFD feeding, pointing to microbiota-mediated effects. In line, the increased abundance of *Erysipelotrichaceae* found in the gut of *Nlrp12*-deficient animals could be associated with the obesity phenotype, whereas supplementation with *Lachnospiraceae*, a bacterial species reduced in abundance in the gut of *Nlrp12* KO animals, or short chain fatty acids (SCFAs) reduced weight gain and inflammation in *Nlrp12*-deficient mice and improved glucose metabolism and insulin sensitivity [115]. So far, genetic alterations of *NLRP12* in humans have not been associated with obesity. However, given the data discussed above, one could think of genetic *NLRP12* variants contributing to weight gain and inflammation via altering the gut microbiome. Further research is needed to shed light on the function of NLRP12 in obesity. 

These data suggest that NLRP12 has a protective role in the development of obesity and IR by influencing gut microbiome composition. 

## 2. Conclusions

The current state of research implicates several NLR proteins to play important roles in obesity and its associated morbidities. Currently available data clearly demonstrate a function of NLRP3, NOD1 and NOD2, and highly suggest an implication of NLRC5 and NLRP12 in adiposity and low-grade inflammation. For NLRP3, NOD1 and NOD2, mechanistic details have been elucidated, primarily in animal models. NOD1 and NLRP3 seem to contribute to the development of obesity, IR and low-grade inflammation, primarily via the induction of pro-inflammatory cytokines, while NOD2 and NLRP12 are thought to oppose metabolic disturbances, NOD2 via the induction of the transcription factor IRF4, and NLRP12 via alteration of the microbiota. However, the triggers for the activation of NLRs in obese individuals are not yet fully understood. Both the classical activation by PAMPs and DAMPs and activation by nutritional compounds like fatty acids are emerging as key players. A central question for the current research is therefore whether the contribution of NLR proteins to obesity-associated low-grade inflammation is “direct”, with higher loads of bacterial substances in the serum of obese individuals resulting in tissue inflammation via NLR-induced pro-inflammatory signaling, or whether they also contribute via other pathways, independent of their function as PRRs. The lack of a well-documented function for NLRC5 as a PRR and the fact that activation of NOD2 by bacterial ligands can reduce low-grade inflammation in high fat intake highly suggest PRR-independent functions for these NLR proteins. 

## 3. Outlook

Based on the data presented above, NLR proteins offer an attractive target for the therapy of obesity and obesity-associated diseases. Antibodies against IL-1β (canakinumab, gevokizumab, LY2189) and IL-1 receptor antagonists (anakinra) represent potential therapeutics for T2DM and atherosclerosis and have already been tested successfully in human studies [85,181,249,250,251,252,253]. In addition, synthetic NOD2 activators such as mifamurtide or inhibitors of RIP2 kinase [226,227] could be used to increase insulin sensitivity [103] or to limit the negative consequences of NOD1 activation, respectively. Elucidating the mechanisms for the contribution of NLRs to obesity and its associated morbidities should therefore be subject to further research. As the role of NLRs as essential players in obesity and its associated morbidities is only beginning to emerge, and associations are so far only known for NOD1, NOD2, NLRP3, NLRP6, NLRP12 and NLRC5, it would be of interest to elucidate a potential role for the other known NLR family members in obesity. Genetic association studies are suitable to shed light on this. Given the plethora of NLR protein functions, some being essentially involved in immune defense against pathogens, the signaling events and molecular mechanisms of NLR signaling in the obesity context need further clarification in order to develop targeted therapeutic options. 

## Figures and Tables

**Figure 1 ijms-24-08595-f001:**
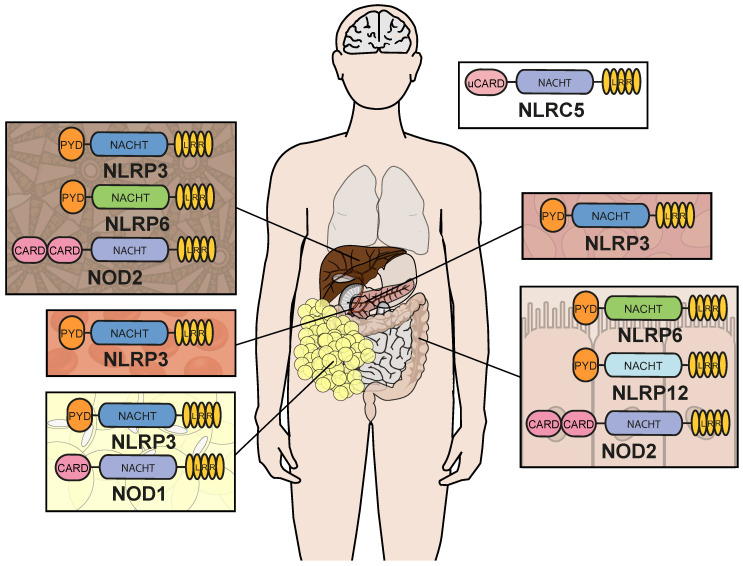
Schematic domain organization and site of action of NLR proteins in obesity and associated morbidities. Shown are the schematic domain organizations of the NLRs discussed in the main text and their main site of action in obesity and its associated morbidities. CARD = caspase activation and recruitment domain, uCARD = untypical CARD, LRR = leucine-rich repeats, NACHT = nucleotide binding and oligomerization domain, PYD = pyrin domain.

**Figure 2 ijms-24-08595-f002:**
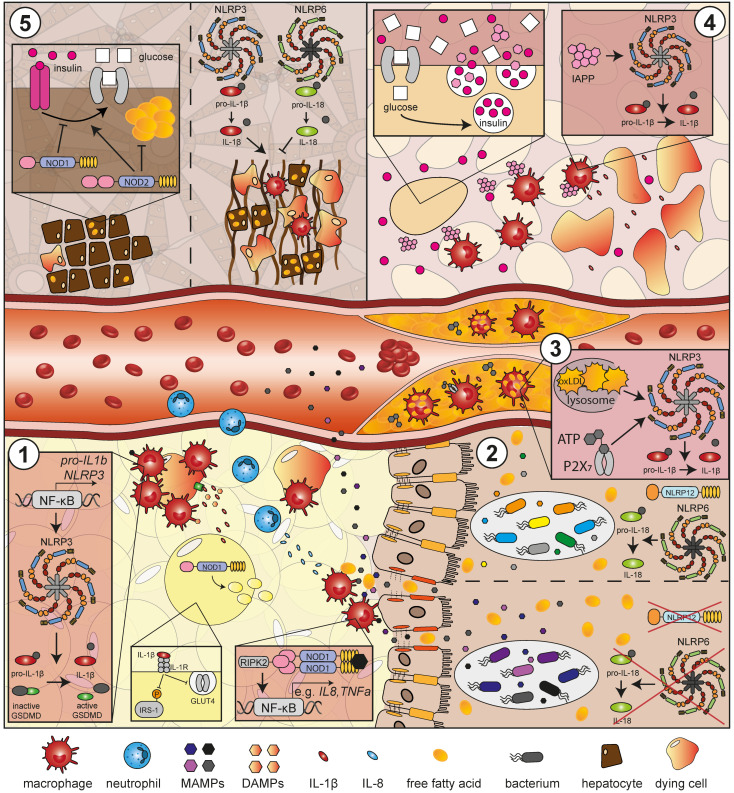
Effects and sites of action of NLR proteins in obesity and associated morbidities. (**1**) In the adipose tissue, metabolic endotoxemia is sensed by adipose tissue macrophages (ATMs) via NOD1. NOD1 activation leads to NF-κB-mediated NLRP3 inflammasome priming but also transcription of pro-inflammatory cytokines, which lead to the recruitment of neutrophils, and adipocyte lipolysis. Activation of the NLRP3 inflammasome induces IL-1β- and Gasdermin D (GSDMD)-mediated AT inflammation, pyroptosis and insulin resistance. (**2**) Deficiency in NLRP6, and subsequently IL-18, or NLRP12 alters commensal gut microbiota composition, leading to increased translocation of microbe-associated molecular patterns (MAMPs) to the circulation (NLRP6) or the induction of obesity, IR and inflammation (NLRP12). (**3**) In atherosclerotic plaques, oxidatively modified low-density lipoprotein cholesterol (oxLDL) crystals lead to lysosomal rupture in macrophages, and release of ATP to activate the NLRP3 inflammasome and subsequently IL-1β release. (**4**) Steadily elevated insulin secretion due to hyperglycemia leads to high secretion and ultimately aggregation of islet amyloid polypeptide (IAPP). IAPP aggregates activate the NLRP3 inflammasome and IL-1β production in pancreatic macrophages, leading to IL-1β-mediated destruction of pancreatic β cells. (**5**) In the liver, IL-18 production via the NLRP6 or NLRP3 inflammasomes leads to decreased liver steatosis and fibrosis, while IL-1β leads to increased liver steatosis and fibrosis. NOD1 reduces while NOD2 increases hepatic insulin sensitivity in response to gut derived activators. Additionally, NOD2 reduces hepatic lipid accumulation.

**Table 1 ijms-24-08595-t001:** Overview concerning the effects of NLRs in obesity and its associated morbidities.

NLR	Model	Effect	Reference
**NLRP3**	mouse	Nlrp3 und Il-1β expression↑in AT in obesity, diabetes and HFD	[58,59,60,61,62]
	in vivo	*Nlrp3^−/−^, Il1r^−/−^:*	
		improvement in blood glucose control, insulin	[63,64,65,66]
		secretion and AT inflammation	
		improvement in liver inflammation and fibrosis	[67]
		*Caspase-1^−/−^*, *Nlrp3^−/−^*, *Asc^−/−^*, *Il1b ^−/−^*, or *Il18 ^−/−^:*	
		number of atherosclerotic lesions ↓	[19,68,69,70,71,72,73]
		**Protective effects**	
		*Il1r^−/−^, Asc^−/−^, Il18^−/−^:*	
		development of obesity, IR and hyperglycemia	[74,75,76,77]
	in vitro	HFD, WD and SFAs trigger Nlrp3 expression and activation	[63,64,65,66]
		IL-1β impairs insulin signal transduction in adipocytes	[62]
		**Protective effects**	
		*Il1b ^−/−^*:	
		postprandial insulin secretion ↓	[34]
	human in vivo	Expression of NLRP3 inflammasome components and IL-1β ↑ in obesity (in AT), atherosclerosis and liver inflammation	[58,67,78,79,80,81,82,83,84]
		Inhibition of IL-1 signaling: improvement of blood glucose levels and systemic inflammation	[85]
		Adipocytes from obese: insulin signal transduction ↓	[62]
	in vitro	NLRP3 activation in monocytes by palmitate	[86,87]
**NLRP6**	mouse in vivo	**Protective effects**	
		*Nlrp6^−/−^*:	
		more severe NAFLD progression	[76]
	mouse in vivo	Hematopoietic *Nod1*^−/−^: pro-inflammatory macrophages in AT ↓ neutrophils in AT, CXCL1 secretion of macrophages ↓ HFD-induced IR ↓ body weight ↔	[7]
		injection of Nod1 ligands: peripheral and hepatic IR ↑	[88]
**NOD1**		lipolysis in white AT ↑	[89]
		After HFD: Nod1 ligands in blood, time-dependent ↑	[7,90]
		*Nod1* in macrophages, skeletal muscle, AT, liver ↑	[90]
		HFD in *Nod1*^−/−^: body weight ↑, resting energy expenditure ↓ pro-inflammatory cells in liver and white AT ↑ HDL ↑, plasma glucose ↓, Glut4 in white AT ↑	[91]
	in vitro	Nod1 activation in 3T3-L1 adipocytes: IR ↑	[88,92]
		inhibition of adipocyte differentiation	[93]
		induction of lipolysis	[89,94,95]
**NOD1**	human in vivo	AT of women with gestational diabetes: *NOD1* expression ↑ GLUT4-dependent glucose uptake ↓	[96]
		Monocytes of T2DM patients: expression of *NOD1* ↑ pro-inflammatory markers on monocytes ↑	[97]
		abdominal subcutaneous AT of MetS patients: *NOD1* expression ↑; correlated with waist circumference, HbA1c, IR and serum levels of IL-6, MCP-1	[98]
	mouse in vivo	*Nod2^−/−^*: body weight gain ↑ visceral AT and fat in liver ↑ IR ↓ inflammation in AT ↑	[99,100,101]
		*Nod2^−/−^* on HFD diet: *Nod2* expression in AT, muscle, liver ↑ bacterial accumulation in the gut ↑, Alpha diversity ↓ bacterial translocation in AT and liver ↑	[100,101]
**NOD2**		Hepatocyte-specific *Nod2^−/−^*: liver inflammation ↑ lipid and cholesterol metabolism ↑ collagen synthesis ↑	[102]
		Nod2 activation in obesity: inflammation in AT ↓ IR ↓ weight or microbiota ↔	[100,103]
		Nod2 activation in non-hematopoietic cells: glucose blood level ↓ pro-inflammatory cytokines ↓	[104]
	in vitro	Nod2 activation in L cells: augmented GLP-1 secretion	[105]
	human in vitro	NOD2 stimulation in adipose-derived adult stem cells: inhibition of adipocyte differentiation	[93]
**NLRC5**	mouse	*Nlrc5^−/−^*:	
	in vivo	amelioration of diabetic nephropathy	[106]
		deterioration of HFD-induced myocardial damage	[107]
		body weight ↑	[107,108]
	human in vivo	Associations of *NLRC5* methylations with obesity and alterations in lipid metabolism	[109,110,111,112,113]
	in vitro	transcriptional regulation of PPARγ target genes by NLRC5	[108,114]
**NLRP12**	mouse in vivo	**Protective effects**	
		*Nlrp12^−/−^*:	
		body weight and AT inflammation ↑ insulin sensitivity ↓	[115]

Overview over the effects of NLRs in obesity and associated morbidities in in vivo and in vitro studies in human and mouse. AT = adipose tissue, GLP-1 = glucagon-like peptide 1, GLUT4 = glucose transporter 4, HDL = high-density lipoprotein cholesterol, HFD = high-fat diet, IR = insulin resistance, NAFLD = non-alcoholic fatty liver disease, MCP-1 = monocyte chemoattractant protein 1, MetS = metabolic syndrome, PPARγ = peroxisome proliferator-activated receptor γ, SFA = saturated fatty acids, T2DM = type 2 diabetes mellitus, WD = Western diet,↑ = increased, ↓ = decreased, ↔ = unchanged.

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
