# Peer review of "NOD-like Receptors—Emerging Links to Obesity and Associated Morbidities"

_ijms, 2023, doi:10.3390/ijms24108595_

Round 1
Reviewer 1 Report
1. Define overweight and obesity (https://www.ncbi.nlm.nih.gov/pmc/articles/PMC8017326/)
2. The definition of MetS is wrong. Authors can use the following citation to describe MetS (https://www.mdpi.com/1660-4601/18/4/1773)
3. Authors need to describe the relation with obesity and its co-morbidities in the form of diagrammatic representation
4. Authors need to cite the human and molecular works documented with Polymorphisms and genes related to NLR
5. Are there any studies documented between NLR group of genes and Obesity or with its co-morbidities?
6. What does mouse model conclude from NLR studies and what do it recommends for the human studies?
7. Knee osteoarthritis has no role with obesity and its co-morbidity or with NLR group of genes
8. Finally, what about cancers? Is there no cancer disease that do play any role with described genes in this review
9. What is the final output of this review
10. What does this study concludes for the human studies?
Author Response
Reviewer 1:
- Define overweight and obesity (https://www.ncbi.nlm.nih.gov/pmc/articles/PMC8017326/)
We thank the reviewer for raising the point of defining the terms ‘overweight’ and ‘obesity’. We agree that overweight and obesity are sometimes defined by percentiles, however this classification to the best of our knowledge is only applied to define overweight and obesity in children and adolescents. As the human data presented in the submitted manuscript was generated in adults only, we would like to stick with our definition of overweight and obesity by BMI (lines 29 – 31). For clarification, we now explicitly state that the BMI is used for classification of body weight in adults. We hope that the reviewer can follow our argumentation.
- The definition of MetS is wrong. Authors can use the following citation to describe MetS (https://www.mdpi.com/1660-4601/18/4/1773)
We agree with the reviewer and apologize for the incomplete definition of the MetS. We now completed the definition by adding the before missing low HDL levels and by further specifying the point of impaired glycaemia.
- Authors need to describe the relation with obesity and its co-morbidities in the form of diagrammatic representation
We thank the reviewer for this comment. We respectfully would like to point out that the effects of the single NLRs in obesity-associated morbidities are already depicted in Figure 2 of our manuscript that covers the most important co-morbidities.
- Authors need to cite the human and molecular works documented with Polymorphisms and genes related to NLR
We thank the reviewer for this suggestion. We now provide information on NLRP3 and IL1B polymorphisms that have been associated with T2DM and obesity (lines 261 – 264). We respectfully would like to point out that for NOD1 and NLRC5, we in the first version of the manuscript already commented on polymorphisms in the respective genes related to obesity (lines 620 and 723 – 726). To the best of our knowledge, for NOD2, NLRP6 and NLRP12 no polymorphisms yet has been described to be associated with obesity. We hope to sufficiently have addressed the reviewers comment.
- Are there any studies documented between NLR group of genes and Obesity or with its co-morbidities?
So far, to the best of our knowledge, only for the six NLR family members discussed in our manuscript robust data on their association with obesity and associated morbidities is known. Thereby, most studies investigating the role of NLR proteins in obesity are primarily focusing on single NLRs or a combination of two functionally related NLR proteins, e.g. NOD1 and NOD2, NLRP3 and NLRP6. Up to date, to the best of our knowledge, there is no study comprehensively investigating the expression of all 22 human NLR family members and their regulation in obesity.
- What does mouse model conclude from NLR studies and what do it recommends for the human studies?
We thank the reviewer for raising this point. In combination with query number #10 in the revised version of our manuscript we now comment on how the data generated in mice could be transferred to humans and which possibilities open up from this transfer. The corresponding paragraphs are highlighted in the revised version.
- Knee osteoarthritis has no role with obesity and its co-morbidity or with NLR group of genes
We do agree with the reviewer, however do wonder why the reviewer raises this point, as we do not comment on knee osteoarthritis in our manuscript.
- Finally, what about cancers? Is there no cancer disease that do play any role with described genes in this review
We thank the reviewer for raising this point. Several NLRs, for example NOD1 and NLRC5, indeed have been implicated in cancer development and promotion. As the main focus of our manuscript however is on obesity and its associated morbidities, we do not comment on the role of NLRs in cancer. It is of course known that obesity is associated with a plethora of cancer entities but to the best of our knowledge, so far there is no data assessing the role of NLRs in obesity-caused cancer. Thus, we refrained from discussing the effects of NLRs in cancer, as we felt this would have been outside the scope of our manuscript. We hope, that the reviewer can follow our argumentation.
- What is the final output of this review
In order to address this point raised by the reviewer, in the revised version of our manuscript restructured our conclusion part, which now gives a short summary of the main text, and added an additional paragraph called ‘outlook’ where we state on present and further work to be done in order to use the data presented in our manuscript for developing treatment strategies. By this, we hope to have sufficiently outlined the final output of our manuscript.
- What does this study concludes for the human studies?
We thank the reviewer for raising this point. In the revised version of our manuscript we now comment on how the data discussed can be transferred to humans and how new treatment strategies could be developed from this. The corresponding paragraphs are highlighted in the revised version.
Reviewer 2 Report
Very extensive article.
With such extensive work, there is no clear division of the article into study, result, conclusion.
Research questions are chaotically arranged, unspecified.
Author Response
Reviewer 2:
- Very extensive article.
We hope that our revisions in response to all reviewed helped to improve the clarity of the article.
- With such extensive work, there is no clear division of the article into study, result, conclusion.
We thank the reviewer for this comment. We respectfully would like to point out that the classical division into study, results and conclusion is not quite applicable to manuscript, given its nature as narrative review. We do however hope to provide structure to the reader by headings and sub-headings and structuring the text into introduction, main part and conclusion. If the reviewer should feel the text insufficiently structured, we could think about introducing additional sub-headings.
- Research questions are chaotically arranged, unspecified.
We thank the reviewer for this comment. Given the nature of our manuscript as narrative review we comment on multiple research aspects on the role of several NLR proteins in obesity and associated morbidities. Different research aspects have been specified in the single paragraphs and structured by using sub-headings. If the reviewer should feel this structure insufficient, we are open for suggestions on how to improve our structure.
Reviewer 3 Report
The manuscript focuses the current state of research on the role of NLR proteins in obesity and the possible mechanisms leading to the NLR activation in the obesity-associated morbidities IR, type 2 diabetes mellitus, atherosclerosis and non-alcoholic fatty liver disease. In addition, the manuscript discuss emerging ideas about possibilities for NLR-based therapeutic interventions of metabolic diseases.
The manuscript is well-organized and the topic is very interesting to the scientific community. However, some problems were found in the manuscript and must be improved.
Kind regards,
English, punctuation and grammar must improved and the following phases need attention:
1. Not only metabolic processes are considerably influenced by the intake of an HFD.
2. It is still unclear which signals mediate the priming and thus drive the expression of 281 NLRP3 and pro-IL1b in the AT in the context of obesity.
3. It is still unclear which signals mediate the priming and thus drive the expression of 281 NLRP3 and pro-IL1b in the AT in the context of obesity.
4. Additionally, ROS have 296 been implicated in inflammasome activation
5. One study failed to show alterations in IR or fat 537 mass in Nod1-/- mice, however in this study mice were only 4 weeks of HFD 183 as compared 538 to 16 weeks 181 and 6 182 weeks feeding duration used in the other experiments.
6. The data discussed here suggest an important role for several members of the NLR 765 protein family in obesity and associated morbidities.
Kind regards,
Author Response
Reviewer 3:
The manuscript focuses the current state of research on the role of NLR proteins in obesity and the possible mechanisms leading to the NLR activation in the obesity-associated morbidities IR, type 2 diabetes mellitus, atherosclerosis and non-alcoholic fatty liver disease. In addition, the manuscript discuss emerging ideas about possibilities for NLR-based therapeutic interventions of metabolic diseases.
The manuscript is well-organized and the topic is very interesting to the scientific community. However, some problems were found in the manuscript and must be improved.
English, punctuation and grammar must improved and the following phases need attention.
We thank the reviewer for the favourable evaluation of our work. To address the reviewer’s comment on the quality of the English language, we re-phrased the sentences highlighted by the reviewer. We marked the revised sentences in the manuscript and hope we could address the reviewer’s concerns.
Round 2
Reviewer 1 Report
The authors have improved but does not followed all the norms. If this time, authors will not justify all the raised comments, then manuscript cannot be considered for the publication. This time be careful in addressing the properly for the raised comments
Author Response
Thank you for your efforts in reviewing the manuscript.
Reviewer 3 Report
Dear Authors:
The manuscript is good for publication in the journal now.
Kind regards,
Author Response

(The authors gave the same response as above.)
